# What Can ResNet Learn Efficiently, Going Beyond Kernels?[*]

**Zeyuan Allen-Zhu**
Microsoft Research AI
zeyuan@csail.mit.edu

**Yuanzhi Li**
Carnegie Mellon University
yuanzhil@andrew.cmu.edu

## Abstract

How can neural networks such as ResNet *efficiently* learn CIFAR-10 with test accuracy more than $96\%$, while other methods, especially kernel methods, fall relatively behind? Can we more provide theoretical justifications for this gap?

Recently, there is an influential line of work relating neural networks to kernels in the over-parameterized regime, proving they can learn certain concept class that is also learnable by kernels with similar test error. Yet, can neural networks provably learn some concept class *better* than kernels?

We answer this positively in the distribution-free setting. We prove neural networks can efficiently learn a notable class of functions, including those defined by three-layer residual networks with smooth activations, without any distributional assumption. At the same time, we prove there are simple functions in this class such that with the same number of training examples, the test error obtained by neural networks can be *much smaller* than *any* kernel method, including neural tangent kernels (NTK).

The main intuition is that *multi-layer* neural networks can implicitly perform hierarchical learning using different layers, which reduces the sample complexity comparing to "one-shot" learning algorithms such as kernel methods. In a follow-up work [2], this theory of hierarchical learning is further strengthened to incorporate the "backward feature correction" process when training deep networks.

In the end, we also prove a computation complexity advantage of ResNet with respect to other learning methods including linear regression over arbitrary feature mappings.

## 1  Introduction

Neural network learning has become a key practical machine learning approach and has achieved remarkable success in a wide range of real-world domains, such as computer vision, speech recognition, and game playing [18, 19, 22, 31]. On the other hand, from a theoretical standpoint, it is less understood that how large-scale, non-convex, non-smooth neural networks can be optimized efficiently over the training data and *generalize* to the test data with *relatively few* training examples.

There has been a sequence of research trying to address this question, showing that under certain conditions neural networks can be learned efficiently [8–10, 15, 16, 21, 24, 25, 32–35, 37, 40]. These provable guarantees typically come with strong assumptions and the proofs heavily rely on them. One common assumption from them is on the *input distribution*, usually being random Gaussian or sufficiently close to Gaussian. While providing great insights to the optimization side of neural networks, it is not clear whether these works emphasizing on Gaussian inputs can coincide with the neural network learning process in practice. Indeed, in nearly all real world data where deep learning is applied to, the input distributions are not close to Gaussians; even worse, there may be

---

[*]Full version and future updates can be found on `https://arxiv.org/abs/1905.10337`.

no simple model to capture such distributions.

The difficulty of modeling real-world distributions brings us back to the traditional PAC-learning language which is *distribution-free*. In this language, one of the most popular, *provable* learning methods is the kernel methods, defined with respect to kernel functions $K(x, x')$ over pairs of data $(x, x')$. The optimization task associated with kernel methods is convex, hence the convergence rate and the generalization error bound are well-established in theory.

Recently, there is a line of work studying the convergence of neural networks in the PAC-learning language, especially for over-parameterized neural networks [1, 3–7, 12–14, 20, 23, 41], putting neural network theory back to the distribution-free setting. Most of these works rely on the so-called Neural Tangent Kernel (NTK) technique [12, 20], by relating the training process of sufficiently over-parameterized (or even infinite-width) neural networks to the learning process over a kernel whose features are defined by the randomly initialized weights of the neural network. In other words, on the same training data set, these works prove that neural networks can efficiently learn a concept class with as good generalization as kernels, but *nothing more* is known.[2]

In contrast, in many practical tasks, neural networks give much better generalization error compared to kernels, although both methods can achieve zero training error. For example, ResNet achieves 96% test accuracy on the CIFAR-10 data set, but NTKs achieve 77% [6] and random feature kernels achieve 85% [29]. This gap becomes larger on more complicated data sets.

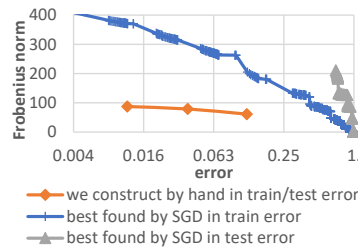

Figure 1: $d = 40, N = 5000$, after exhaustive search in network size, learning rate, weight decay, randomly initialized SGD still cannot find solutions with Frobenius norm comparable to what we construct by hand. Details and more experiments in Section 9.2.

To separate the generalization power of neural networks from kernel methods, the recent work [36] tries to identify conditions where the solutions found by neural networks provably generalize better than kernels. This approach assumes that the optimization converges to minimal complexity solutions (i.e. the ones minimizing the value of the regularizer, usually the sum of squared Frobenius norms of weight matrices) of the training objective. However, for most practical applications, it is unclear how, when training neural networks, minimal complexity solutions can be found efficiently by local search algorithms such as stochastic gradient descent. In fact, it is not true even for rather simple problems (see Figure 1).[3] Towards this end, the following fundamental question is largely unsolved:

> *Can neural networks efficiently and distribution-freely learn a concept class,*
>
> *with better generalization than kernel methods?*

In this paper, we give arguably the *first* positive answer to this question for neural networks with ReLU activations. We show without any distributional assumption, a three-layer residual network (ResNet) can (improperly) learn a concept class that includes three-layer ResNets of smaller size and smooth activations. This learning process can be efficiently done by stochastic gradient descent (SGD), and the generalization error is also small if polynomially many training examples are given.

More importantly, we give a *provable separation* between the generalization error obtained by neural networks and kernel methods. For some $\delta \in (0, 1)$, with $N = O(\delta^{-2})$ training samples, we prove that neural networks can *efficiently* achieve generalization error $\delta$ for this concept class over *any distribution*; in contrast, there exists rather simple distributions such that any kernel method (in-

cluding NTK, recursive kernel, etc) cannot have generalization error better than $\sqrt{\delta}$ for this class. To the best of our knowledge, this is the first work that gives provable, *efficiently achievable* separation between neural networks with ReLU activations and kernels in the distribution-free setting. In the end, we also prove a computation complexity advantage of neural networks with respect to linear regression over arbitrary feature mappings as well.

**Roadmap.** We present detailed overview of our positive and negative results in Section 2 and 3. Then, we introduce notations in Section 4, formally define our concept class in Section 5, and give proof overviews in Section 6 and 8.

## 2 Positive Result: The Learnability of Three-Layer ResNet

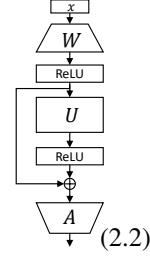

In this paper, we consider *learner networks* that are single-skip three-layer ResNet with ReLU activation, defined as a function $\mathsf{out}\colon \mathbb{R}^d \to \mathbb{R}^k$:

$$\mathsf{out}(x) = \mathbf{A}\left(\sigma\left(\mathbf{W}x + b_1\right) + \sigma\left(\mathbf{U}\sigma\left(\mathbf{W}x + b_1\right) + b_2\right)\right) \qquad (2.1)$$

Here, $\sigma$ is the ReLU function, $\mathbf{W} \in \mathbb{R}^{m \times d}$ and $\mathbf{U} \in \mathbb{R}^{m \times m}$ are the hidden weights, $\mathbf{A} \in \mathbb{R}^{k \times m}$ is the output weight, and $b_1, b_2 \in \mathbb{R}^m$ are two bias vectors.

We wish to learn a concept class given by target functions that can be written as

$$\mathcal{H}(x) = \mathcal{F}(x) + \alpha\mathcal{G}\left(\mathcal{F}(x)\right) \qquad (2.2)$$

where $\alpha \in [0, 1)$ and $\mathcal{G}\colon \mathbb{R}^k \to \mathbb{R}^k, \mathcal{F}\colon \mathbb{R}^d \to \mathbb{R}^k$ are two functions that can be written as two-layer networks with smooth activations (see Section 5 for the formal definition). Intuitively, the target function is a mixture of two parts: the base signal $\mathcal{F}$, which is simpler and contributes more to the target, and the composite signal $\mathcal{G}(\mathcal{F})$, which is more complicated but contributes less. As an analogy, $\mathcal{F}$ could capture the signal in which "85%" examples in CIFAR-10 can be learned by kernel methods, and $\mathcal{G}(\mathcal{F})$ could capture the additional "11%" examples that are more complicated.

The goal is to use three-layer ResNet (2.1) to *improperly* learn this concept class (2.2), meaning learning "both" the base and composite signals, with as few samples as possible. In this paper, we consider a simple $\ell_2$ regression task where the features $x \in \mathbb{R}^d$ and labels $y \in \mathbb{R}^k$ are sampled from some unknown distribution $\mathcal{D}$. Thus, given a network $\mathsf{out}(x)$, the population risk is

$$\mathbb{E}_{(x,y)\sim\mathcal{D}} \frac{1}{2} \left\|\mathsf{out}(x) - y\right\|_2^2 \ .$$

To illustrate our result, we first assume *for simplicity* that $y = \mathcal{H}(x)$ for some $\mathcal{H}$ of the form (2.2) (so the optimal target has zero regression error). Our main theorem can be sketched as follows.

Let $C_{\mathcal{F}}$ and $C_{\mathcal{G}}$ respectively be the individual "complexity" of $\mathcal{F}$ and $\mathcal{G}$, which at a high level, capture the size and smoothness of $\mathcal{F}$ and $\mathcal{G}$. This complexity notion shall be formally introduced in Section 4, and is used by prior works such as [3, 7, 39].

**Theorem 1** (ResNet, sketched). *For any distribution over $x$, for every $\delta \in \left((\alpha C_{\mathcal{G}})^4, 1\right)$, with probability at least $0.99$, SGD efficiently learns a network $\mathsf{out}(x)$ in the form (2.1) satisfying*

$$\mathbb{E}_{(x,y)\sim\mathcal{D}} \frac{1}{2} \left\|\mathsf{out}(x) - y\right\|_2^2 \leq \delta \quad using \quad N = \widetilde{O}\Big(\frac{C_{\mathcal{F}}^2}{\delta^2}\Big) \ samples$$

*The running time of SGD is polynomial in $\mathsf{poly}(C_{\mathcal{G}}, C_{\mathcal{F}}, \alpha^{-1})$.*

In other words, ResNet is capable of achieving population risk $\alpha^4$, or equivalently learning the output $\mathcal{H}(x)$ up to $\alpha^2$ error. In our full theorem, we also allow label $y$ to be generated from $\mathcal{H}(x)$ with error, thus our result also holds in the agnostic learning framework.

### 2.1 Our Contributions

Our main contribution is to obtain time and sample complexity in $C_{\mathcal{F}}$ and $C_{\mathcal{G}}$ *without any dependency* on the composed function $\mathcal{G}(\mathcal{F})$ as in prior work [3, 39]. We illustrate this crucial difference with an example. Suppose $x \sim \mathcal{N}(0, \mathbf{I}/d)$, $k = 2$ and $\mathcal{F} \in \mathbb{R}^d \to \mathbb{R}^2$ consists of two linear function: $\mathcal{F}(x) = \left(\langle w_1^*, x\rangle, \langle w_2^*, x\rangle\right)$ with $\|w_1^*\|_2, \|w_2^*\|_2 = \sqrt{d}$, and $\mathcal{G}$ is degree-10 polynomial with constant coefficient. As we shall see, $C_{\mathcal{F}} = O(\sqrt{d})$ and $C_{\mathcal{G}} = \widetilde{O}(1)$. Theorem 1 implies

- we need $\widetilde{O}(d)$ samples to efficiently learn $\mathcal{H} = \mathcal{F} + \alpha\mathcal{G}(\mathcal{F})$ up to accuracy $\widetilde{O}(\alpha^2)$.

In contrast, the complexity of $\mathcal{G}(\mathcal{F})$ is $\widetilde{O}((\sqrt{d})^{10})$, so

- prior works [3, 39] need $\widetilde{\Omega}(d^{10})$ samples to *efficiently* learn $\mathcal{H}$ up to any accuracy $o(\alpha)$,

even if $\mathcal{G}(x)$ is of some simple form such as $\langle w_1^*, x\rangle^{10} - \langle w_2^*, x\rangle^{10}$.[4]

**Inductive Bias.** Our network is over-parameterized, thus intuitively in the example above, with only $O(d)$ training examples, the learner network could over-fit to the training data since it has to decide from a set of $d^{10}$ many possible coefficients to learn the degree 10 polynomial $\mathcal{G}$. This is indeed the case if we learn the target function using kernels, or possibly even learn it with a two-layer network. However, three-layer ResNet posts a *completely different* inductive bias, and manages to avoid over-fitting to $\mathcal{G}(\mathcal{F})$ with the help from $\mathcal{F}$.

**Implicit Hierarchical Learning.** Since $\mathcal{H}(x) = \mathcal{F}(x) + \alpha \mathcal{G}(\mathcal{F}(x))$, if we only learn $\mathcal{F}$ but not $\mathcal{G}(\mathcal{F})$, we will have regression error $\approx (\alpha C_{\mathcal{G}})^2$. Thus, to get to regression error $(\alpha C_{\mathcal{G}})^4$, Theorem 1 shows that ResNet is also capable of learning $\mathcal{G}(\mathcal{F})$ up to some good accuracy with *relatively few* training examples. This is also observed in practice, where with this number of training examples, three-layer fully-connected networks and kernel methods can indeed fail to learn $\mathcal{G}(\mathcal{F})$ up to any non-trivial accuracy, see Figure 2.

Intuitively, there is a hierarchy of the learning process: we would like to first learn $\mathcal{F}$, and then we could learn $\mathcal{G}(\mathcal{F})$ much easier with the help of $\mathcal{F}$ using the *residual link*. In our learner network (2.1), the first hidden layer serves to learn $\mathcal{F}$ and the second hidden layer serves to learn $\mathcal{G}$ with the help of $\mathcal{F}$, which reduces the sample complexity. However, the important message is that $\mathcal{F}$ and $\mathcal{G}$ are *not* given as separate data to the network, rather the learning algorithm has to *disentangle* them from the "combined" function $\mathcal{H} = \mathcal{F} + \alpha \mathcal{G}(\mathcal{F})$ automatically during the training process. Moreover, since we *train both layers simultaneously*, the learning algorithm also has to *distribute* the learning task of $\mathcal{F}$ and $\mathcal{G}$ onto different layers automatically.

We also emphasize that our result cannot be obtained by layer-wise training of the ResNet, that is, first training the hidden layer close to the input, and then training the hidden layer close to the output. Since it could be the case the first layer incurs some $\alpha$ error (since it cannot learn $\mathcal{G}(\mathcal{F})$ directly), then it could be really hard, or perhaps impossible, for the second layer to fix it only using inputs of the form $\mathcal{F}(x) \pm \alpha$. In other words, it is crucial that the two hidden layers are *simultaneously trained*. [5]

**A follow-up work.** In a follow-up work [2], this theory of hierarchical learning is *significantly strengthened* to further incorporate the "backward feature correction" when training deep neural networks. In the language of this paper, when the two layers trained together, given enough samples, the accuracy in the first layer can actually be *improved* from $\mathcal{F} \pm \alpha$ to *arbitrarily close to* $\mathcal{F}$ during the training process. As a consequence, the final training and generalization error can be arbitrarily small as well, as opposite to $\alpha^4$ in this work. The new "backward feature correction" is also critical to extend the hierarchical learning process from 3 layers to *arbitrarily number of layers*.

## 3 Negative Results

### 3.1 Limitation of Kernel Methods

Given (Mercer) kernels $K_1, \ldots, K_k : \mathbb{R}^{d \times d} \to \mathbb{R}$ and training examples $\{(x^{(i)}, y^{(i)})\}_{i \in [N]}$ from $\mathcal{D}$, a kernel method tries to learn a function $\mathfrak{K} : \mathbb{R}^d \to \mathbb{R}^k$ where each

$$\mathfrak{K}_j(x) = \sum_{n \in [N]} K_j(x, x^{(n)}) \cdot w_{j,n} \tag{3.1}$$

is parameterized by a weight vector $w_j \in \mathbb{R}^N$. Usually, for the $\ell_2$ regression task, a kernel method finds the optimal weights $w_1, \ldots, w_k \in \mathbb{R}^N$ by solving the following convex minimization problem

$$\frac{1}{N} \sum_{i=1}^{N} \sum_{j \in [k]} \left( \sum_{n \in [N]} K_j(x^{(i)}, x^{(n)}) w_{j,n} - y_j^{(i)} \right)^2 + R(w) \qquad (3.2)$$

for some convex regularizer $R(w)$.[6] In this paper, however, we do not make assumptions about how $\mathfrak{K}(x)$ is found as the optimal solution of the training objective. Instead, we focus on *any* kernel regression function that can be written in the form (3.1).

Most of the widely-used kernels are Mercer kernels.[7] This includes (1) *Gaussian kernel* $K(x,y) = e^{-\|x-y\|_2^2/h}$; (2) *arcsin kernel* $K(x,y) = \arcsin\left(\langle x, y\rangle/(\|x\|_2\|y\|_2)\right)$; (3) *recursive kernel* with any recursive function [39]; (4) *random feature kernel* $K(x,y) = \mathbb{E}_{w \sim \mathcal{W}} \phi_w(x)\phi_w(y)$ for any function $\phi_w(\cdot)$ and distribution $\mathcal{W}$; (5) the *conjugate kernel* defined by the last hidden layer of random initialized neural networks [11]; (6) the *neural tangent kernels (NTK)* for fully-connected [20] networks, convolutional networks [6, 38] or more generally for any architectures [38].

Our theorem can be sketched as follows:

**Theorem 3** (kernel, sketched). *For every constant $k \geq 2$, for every sufficiently large $d \geq 2$, there exist concept classes consisting of functions $\mathcal{H}(x) = \mathcal{F}(x) + \alpha\mathcal{G}(\mathcal{F}(x))$ with complexities $C_\mathcal{F}, C_\mathcal{G}$ and $\alpha \in (0, \frac{1}{C_\mathcal{G}})$ such that, letting*

$N_{\mathsf{res}}$ *be the sample complexity from Theorem 1 to achieve $\alpha^{3.9}$ population risk,*

*then there exists simple distributions $\mathcal{D}$ over $(x, \mathcal{H}(x))$ such that, for at least 99% of the functions $\mathcal{H}$ in this concept class, even given $N = O\left((N_{\mathsf{res}})^{k/2}\right)$ training samples from $\mathcal{D}$, any function $\mathfrak{K}(x)$ of the form (3.1) has to suffer population risk*

$$\mathbb{E}_{(x,y)\sim\mathcal{D}} \frac{1}{2} \|\mathfrak{K}(x) - y\|_2^2 > \alpha^2 \quad \textit{even if the label } y = \mathcal{H}(x) \textit{ has no error.}$$

**Contribution and Intuition.** Let us compare this to Theorem 1. While both algorithms are efficient, neural networks (trained by SGD) achieve population risk $\alpha^{3.9}$ using $N_{\mathsf{res}}$ samples for *any* distribution over $x$, while kernel methods cannot achieve any population risk better than $\alpha^2$ for some simple distributions even with $N = (N_{\mathsf{res}})^{k/2} \gg N_{\mathsf{res}}$ samples.[8] Our two theorems together gives a provable separation between the generalization error of the solutions found by neural networks and kernel methods, in the *efficiently computable regime*.

More specifically, recall $C_\mathcal{F}$ and $C_\mathcal{G}$ only depend on individual complexity of $\mathcal{G}, \mathcal{F}$, but not on $\mathcal{G}(\mathcal{F})$. In Theorem 3, we will construct $\mathcal{F}$ as linear functions and $\mathcal{G}$ as degree-$k$ polynomials. This ensures $C_\mathcal{F} = O(\sqrt{d})$ and $C_\mathcal{G} = O(1)$ for $k$ being constant, but the combined complexity of $\mathcal{G}(\mathcal{F})$ is as high as $\Omega(d^{k/2})$. Since ResNet can perform hierarchical learning, it only needs sample complexity $N_{\mathsf{res}} = O(d/\alpha^8)$ instead of paying (square of) the combined complexity $\Omega(d^k)$.

In contrast, a kernel method is not hierarchical: rather than discovering $\mathcal{F}$ first and then learning $\mathcal{G}(\mathcal{F})$ with the guidance of it, kernel method tries to learn everything *in one shot*. This unavoidably requires the sample complexity to be at least $\Omega(d^k)$. Intuitively, as the kernel method tries to learn $\mathcal{G}(\mathcal{F})$ from scratch, this means that it has to take into account all $\Omega(d^k)$ many possible choices of $\mathcal{G}(\mathcal{F})$ (recall that $\mathcal{G}$ is a degree $k$ polynomial over dimension $d$). On the other hand, a kernel method with $N$ samples only has $N$-degrees of freedom (for each output dimension). This means, if $N \ll o(d^k)$, kernel method simply does not have enough degrees of freedom to distinguish between different $\mathcal{G}(\mathcal{F})$, so has to pay $\Omega(\alpha^2)$ in population risk. Choosing for instance $\alpha = d^{-0.1}$, we have the desired negative result for all $N \leq O\left((N_{\mathsf{res}})^{k/2}\right) \ll o(d^k)$.

## 3.2 Limitation of Linear Regression Over Feature Mappings

Given an arbitrary feature mapping $\phi\colon \mathbb{R}^d \to \mathbb{R}^D$, one may consider learning a linear function over $\phi$. Namely, to learn a function $\mathfrak{F}\colon \mathbb{R}^d \to \mathbb{R}^k$ where each

$$\mathfrak{F}_j(x) = w_j^\top \phi(x) \tag{3.3}$$

is parameterized by a weight vector $w_j \in \mathbb{R}^D$. Usually, these weights are determined by minimizing the following regression objective:[9]

$$\frac{1}{N} \sum_{i\in[N]} \sum_{j\in[k]} \left( w_j^\top \phi(x^{(i)}) - y_j^{(i)} \right)^2 + R(w)$$

for some regularizer $R(w)$. In this paper, we do not make assumptions about how the weighted are found. Instead, we focus on *any* linear function over such feature mapping in the form (3.3).

**Theorem 4** (feature mapping, sketched). *For sufficiently large integers $d, k$, there exist concept classes consisting of functions $\mathcal{H}(x) = \mathcal{F}(x) + \alpha \mathcal{G}\left(\mathcal{F}(x)\right)$ with complexities $C_{\mathcal{F}}, C_{\mathcal{G}}$ and $\alpha \in (0, \frac{1}{C_{\mathcal{G}}})$ such that, letting*

$T_{\mathsf{res}}$ *be the time complexity from Theorem 1 to achieve $\alpha^{3.9}$ population risk,*

*then for at least $99\%$ of the functions $\mathcal{H}$ in this concept class, even with arbitrary $D = (T_{\mathsf{res}})^2$ dimensional feature mapping, any function $\mathfrak{F}(x)$ of the form (3.3) has to suffer population risk*

$$\mathbb{E}_{(x,y)\sim\mathcal{D}} \frac{1}{2} \|\mathfrak{F}(x) - y\|_2^2 > \alpha^2 \quad \text{even if the label } y = \mathcal{H}(x) \text{ has no error.}$$

**Interpretation.** Since any algorithm that optimizes linear functions over $D$-dimensional feature mapping has to run in time $\Omega(D)$, this proves a time complexity separation between neural networks (say, for achieving population risk $\alpha^{3.9}$) and linear regression over feature mappings (for achieving even any population risk better than $\alpha^2 \gg \alpha^{3.9}$). Usually, such an algorithm also has to suffer from $\Omega(D)$ space complexity. If that happens, we also have a space complexity separation. Our hard instance in proving Theorem 4 is the same as Theorem 3, and the proof is analogous.

## 4 Notations

We denote by $\|w\|_2$ and $\|w\|_\infty$ the Euclidean and infinity norms of vectors $w$, and $\|w\|_0$ the number of non-zeros of $w$. We also abbreviate $\|w\| = \|w\|_2$ when it is clear from the context. We denote the row $\ell_p$ norm for $\mathbf{W} \in \mathbb{R}^{m\times d}$ (for $p \geq 1$) as

$$\|\mathbf{W}\|_{2,p} := \left( \sum_{i\in[m]} \|w_i\|_2^p \right)^{1/p}.$$

By definition, $\|\mathbf{W}\|_{2,2} = \|\mathbf{W}\|_F$ is the Frobenius norm of $\mathbf{W}$. We use $\|\mathbf{W}\|_2$ to denote the matrix spectral norm. For a diagonal matrix $D$ we use $\|D\|_0$ to denote its sparsity. For a matrix $\mathbf{W} \in \mathbb{R}^{m\times d}$, we use $\mathbf{W}_i$ or $w_i$ to denote the $i$-th row of $\mathbf{W}$.

We use $\mathcal{N}(\mu, \sigma)$ to denote Gaussian distribution with mean $\mu$ and variance $\sigma$; or $\mathcal{N}(\mu, \Sigma)$ to denote Gaussian vector with mean $\mu$ and covariance $\Sigma$. We use $\mathbb{1}_{event}$ or $\mathbb{1}[event]$ to denote the indicator function of whether $event$ is true. We use $\sigma(\cdot)$ to denote the ReLU function, namely $\sigma(x) = \max\{x, 0\} = \mathbb{1}_{x\geq 0} \cdot x$. Given univariate function $f\colon \mathbb{R} \to \mathbb{R}$, we also use $f$ to denote the same function over vectors: $f(x) = (f(x_1), \dots, f(x_m))$ if $x \in \mathbb{R}^m$.

For notation simplicity, throughout this paper "with high probability" (or w.h.p.) means with probability $1 - e^{-c\log^2 m}$ for a sufficiently large constant $c$. We use $\widetilde{O}$ to hide $\mathsf{polylog}(m)$ factors.

**Function complexity.** The following notions introduced in [3] measure the complexity of any infinite-order smooth function $\phi\colon \mathbb{R} \to \mathbb{R}$. Suppose $\phi(z) = \sum_{i=0}^\infty c_i z^i$ is its Taylor expansion.

$$\mathfrak{C}_\varepsilon(\phi) = \mathfrak{C}_\varepsilon(\phi, 1) := \sum_{i=0}^\infty \left( (C^*)^i + \left( \frac{\sqrt{\log(1/\varepsilon)}}{\sqrt{i}} C^* \right)^i \right) |c_i|$$

$$\mathfrak{C}_{\mathfrak{s}}(\phi) = \mathfrak{C}_{\mathfrak{s}}(\phi, 1) := C^* \sum_{i=0}^\infty (i+1) |c_i|$$

where $C^*$ is a sufficiently large constant (e.g., $10^4$).

*Example* 4.1. If $\phi(z) = e^{c\cdot z} - 1$, $\sin(c\cdot z)$, $\cos(c\cdot z)$ or degree-$c$ polynomial for constant $c$, then $\mathfrak{C}_\varepsilon(\phi, 1) = o(1/\varepsilon)$ and $\mathfrak{C}_{\mathfrak{s}}(\phi, 1) = O(1)$. If $\phi(z) = \mathrm{sigmoid}(z)$ or $\tanh(z)$, to get $\varepsilon$ approximation

we can truncate their Taylor series at degree $\Theta(\log \frac{1}{\varepsilon})$. One can verify that $\mathfrak{C}_\varepsilon(\phi, 1) \leq \mathsf{poly}(1/\varepsilon)$ by the fact that $(\log(1/\varepsilon)/i)^i \leq \mathsf{poly}(\varepsilon^{-1})$ for every $i \geq 1$, and $\mathfrak{C}_\mathfrak{s}(\phi, 1) \leq O(1)$.

# 5 Concept Class

We consider learning some unknown distribution $\mathcal{D}$ of data points $z = (x, y) \in \mathbb{R}^d \times \mathbb{R}^k$, where $x \in \mathbb{R}^d$ is the input vector and $y$ is the associated label. Let us consider *target functions* $\mathcal{H} \colon \mathbb{R}^d \to \mathbb{R}^k$ coming from the following concept class.

**Concept 1.** *$\mathcal{H}$ is given by two smooth functions $\mathcal{F}, \mathcal{G} : \mathbb{R}^k \to \mathbb{R}^k$ and a value $\alpha \in \mathbb{R}_+$:*

$$\mathcal{H}(x) = \mathcal{F}(x) + \alpha \mathcal{G}\left(\mathcal{F}(x)\right) \quad , \tag{5.1}$$

*where for each output coordinate $r$,*

$$\mathcal{F}_r(x) = \sum_{i \in [p_\mathcal{F}]} a^*_{\mathcal{F},r,i} \cdot \mathcal{F}_{r,i}\left(\langle w^*_{r,i}, x \rangle\right) \quad and \quad \mathcal{G}_r(h) = \sum_{i \in [p_\mathcal{G}]} a^*_{\mathcal{G},r,i} \cdot \mathcal{G}_{r,i}\left(\langle v^*_{r,i}, h \rangle\right) \tag{5.2}$$

*for some parameters $a^*_{\mathcal{F},r,i}, a^*_{\mathcal{G},r,i} \in [-1, 1]$ and vectors $w^*_{r,i} \in \mathbb{R}^d$ and $v^*_{r,i} \in \mathbb{R}^k$. We assume for simplicity $\|w^*_{r,i}\|_2 = \|v^*_{r,i}\|_2 = 1/\sqrt{2}$.[10] For simplicity, we assume $\|x\|_2 = 1$ and $\|\mathcal{F}(x)\|_2 = 1$ for $(x, y) \sim \mathcal{D}$ and in Appendix A we state a more general Concept 2 without these assumptions.[11]*

We denote by $\mathfrak{C}_\varepsilon(\mathcal{F}) = \max_{r,i}\{\mathfrak{C}_\varepsilon(\mathcal{F}_{r,i})\}$ and $\mathfrak{C}_\mathfrak{s}(\mathcal{F}) = \max_{r,i}\{\mathfrak{C}_\mathfrak{s}(\mathcal{F}_{r,i})\}$. Intuitively, $\mathcal{F}$ and $\mathcal{G}$ are both generated by two-layer neural networks with smooth activation functions $\mathcal{F}_{r,i}$ and $\mathcal{G}_{r,i}$.

Borrowing the *agnostic PAC-learning language*, our concept class consists of all functions $\mathcal{H}(x)$ in the form of Concept 1 with complexity bounded by tuple $(p_F, C_F, p_G, C_G)$. Let OPT be the population risk achieved by the *best* target function in this concept class. Then, our goal is to learn this concept class with population risk $O(\mathsf{OPT}) + \varepsilon$ using sample and time complexity *polynomial* in $p_F, C_F, p_G, C_G$ and $1/\varepsilon$. In the remainder of this paper, to simplify notations, we do not explicitly define this concept class parameterized by $(p_F, C_F, p_G, C_G)$. Instead, we equivalently state our theorem with respect to any (unknown) fixed target function $\mathcal{H}$ with with population risk OPT:

$$\mathbb{E}_{(x,y)\sim\mathcal{D}}\left[\tfrac{1}{2}\|\mathcal{H}(x) - y\|_2^2\right] \leq \mathsf{OPT} \quad .$$

In the analysis we adopt the following notations. For every $(x, y) \sim \mathcal{D}$, it satisfies $\|\mathcal{F}(x)\|_2 \leq \mathfrak{B}_\mathcal{F}$ and $\|\mathcal{G}(\mathcal{F}(x))\|_2 \leq \mathfrak{B}_{\mathcal{F}\circ\mathcal{G}}$. We assume $\mathcal{G}(\cdot)$ is $\mathfrak{L}_\mathcal{G}$-Lipschitz continuous. It is a simple exercise (see Fact A.3) to verify that $\mathfrak{L}_\mathcal{G} \leq \sqrt{k}p_\mathcal{G}\mathfrak{C}_\mathfrak{s}(\mathcal{G})$, $\mathfrak{B}_\mathcal{F} \leq \sqrt{k}p_\mathcal{F}\mathfrak{C}_\mathfrak{s}(\mathcal{F})$ and $\mathfrak{B}_{\mathcal{F}\circ\mathcal{G}} \leq \mathfrak{L}_\mathcal{G}\mathfrak{B}_\mathcal{F} + \sqrt{k}p_\mathcal{G}\mathfrak{C}(\mathcal{G}) \leq kp_\mathcal{F}\mathfrak{C}_\mathfrak{s}(\mathcal{F})p_\mathcal{G}\mathfrak{C}_\mathfrak{s}(\mathcal{G})$.

# 6 Overview of Theorem 1

We learn the unknown distribution $\mathcal{D}$ with three-layer ResNet with ReLU activation (2.1) as learners. For notation simplicity, we absorb the bias vector into weight matrix: that is, given $\mathbf{W} \in \mathbb{R}^{m \times d}$ and bias $b_1 \in \mathbb{R}^m$, we rewrite $\mathbf{W}x + b$ as $\mathbf{W}(x, 1)$ for a new weight matrix $\mathbf{W} \in \mathbb{R}^{m \times (d+1)}$. We also re-parameterize $\mathbf{U}$ as $\mathbf{U} = \mathbf{VA}$ and we find this parameterization (similar to the "bottleneck" structure in ResNet) simplifies the proof and also works well empirically for our concept class. After such notation simplification and re-parameterization, we can rewrite $\mathsf{out}(x) \colon \mathbb{R}^d \to \mathbb{R}^k$ as

$$\mathsf{out}(\mathbf{W}, \mathbf{V}; x) = \mathsf{out}(x) = \mathsf{out}_1(x) + \mathbf{A}\sigma\left((\mathbf{V}^{(0)} + \mathbf{V})(\mathsf{out}_1(x), 1)\right)$$

$$\mathsf{out}_1(\mathbf{W}, \mathbf{V}; x) = \mathsf{out}_1(x) = \mathbf{A}\sigma(\mathbf{W}^{(0)} + \mathbf{W})(x, 1) \quad .$$

Above, $\mathbf{A} \in \mathbb{R}^{k \times m}, \mathbf{V}^{(0)} \in \mathbb{R}^{m \times (k+1)}, \mathbf{W}^{(0)} \in \mathbb{R}^{m \times (d+1)}$ are weight matrices corresponding to random initialization, and $\mathbf{W} \in \mathbb{R}^{m \times (k+1)}, \mathbf{W} \in \mathbb{R}^{m \times (d+1)}$ are the additional weights to be learned by the algorithm. To prove the strongest result, we only train $\mathbf{W}, \mathbf{V}$ and do not train $\mathbf{A}$.[12]

We consider random Gaussian initialization where the entries of $\mathbf{A}, \mathbf{W}^{(0)}, \mathbf{V}^{(0)}$ are independently generated as follows:

$$\mathbf{A}_{i,j} \sim \mathcal{N}\left(0, \tfrac{1}{m}\right) \qquad [\mathbf{W}^{(0)}]_{i,j} \sim \mathcal{N}\left(0, \sigma_w^2\right) \qquad [\mathbf{V}^{(0)}]_{i,j} \sim \mathcal{N}\left(0, \sigma_v^2/m\right)$$

In this paper we focus on the $\ell_2$ loss function between $\mathcal{H}$ and out, given as:

$$\mathsf{Obj}(\mathbf{W}, \mathbf{V}; (x, y)) = \frac{1}{2}\|y - \mathsf{out}(\mathbf{W}, \mathbf{V}; x)\|_2^2 \tag{6.1}$$

We consider the vanilla SGD algorithm. Starting from $\mathbf{W}_0 = \mathbf{0}, \mathbf{V}_0 = \mathbf{0}$, in each iteration $t = 0, 1, \ldots, T-1$, it receives a random sample $(x_t, y_t) \sim \mathcal{D}$ and performs SGD updates[13]

$$\mathbf{W}_{t+1} \leftarrow \mathbf{W}_t - \eta_w \tfrac{\partial \mathsf{Obj}(\mathbf{W}, \mathbf{V}; (x_t, y_t))}{\partial \mathbf{W}}\big|_{\mathbf{W}=\mathbf{W}_t, \mathbf{V}=\mathbf{V}_t}$$

$$\mathbf{V}_{t+1} \leftarrow \mathbf{V}_t - \eta_v \tfrac{\partial \mathsf{Obj}(\mathbf{W}, \mathbf{V}; (x_t, y_t))}{\partial \mathbf{V}}\big|_{\mathbf{W}=\mathbf{W}_t, \mathbf{V}=\mathbf{V}_t}$$

**Theorem 1.** *Under Concept 1 or Concept 2, for every $\alpha \in \left(0, \widetilde{\Theta}(\frac{1}{kp_{\mathcal{G}}\mathfrak{C}_{\mathfrak{s}}(\mathcal{G})})\right)$ and $\delta \geq \mathsf{OPT} + \widetilde{\Theta}\left(\alpha^4(kp_{\mathcal{G}}\mathfrak{C}_{\mathfrak{s}}(\mathcal{G}))^4(1 + \mathfrak{B}_{\mathcal{F}})^2\right)$. There exist $M = \mathsf{poly}(\mathfrak{C}_\alpha(\mathcal{F}), \mathfrak{C}_\alpha(\mathcal{G}), p_{\mathcal{F}}, \alpha^{-1})$ satisfying that for every $m \geq M$, with high probability over $\mathbf{A}, \mathbf{W}^{(0)}, \mathbf{V}^{(0)}$, for a wide range of random initialization parameters $\sigma_w, \sigma_v$ (see Table 1), choosing*

$$T = \widetilde{\Theta}\left(\tfrac{(kp_{\mathcal{F}}\mathfrak{C}_{\mathfrak{s}}(\mathcal{F}))^2}{\min\{1, \delta^2\}}\right) \quad \eta_w = \widetilde{\Theta}\left(\min\{1, \delta\}\right) \quad \eta_v = \eta_w \cdot \widetilde{\Theta}\left(\tfrac{\alpha p_{\mathcal{G}}\mathfrak{C}_{\mathfrak{s}}(\mathcal{G})}{p_{\mathcal{F}}\mathfrak{C}_{\mathfrak{s}}(\mathcal{F})}\right)^2$$

*With high probability, the SGD algorithm satisfies*

$$\tfrac{1}{T}\sum_{t=0}^{T-1}\mathbb{E}_{(x,y)\sim\mathcal{D}}\|\mathcal{H}(x) - \mathsf{out}(\mathbf{W}_t, \mathbf{V}_t; x)\|_2^2 \leq O(\delta) \ .$$

As a corollary, under Concept 1, we can archive population risk

$$O(\mathsf{OPT}) + \widetilde{\Theta}\left(\alpha^4(kp_{\mathcal{G}}\mathfrak{C}_{\mathfrak{s}}(\mathcal{G}))^4\right) \quad \text{using sample complexity } T \ . \tag{6.2}$$

*Remark* 6.1. Our Theorem 1 is almost in the PAC-learning language, except that the final error has an additive $\alpha^4$ term that can not be arbitrarily small.

## 6.1  Proof Overview

In the analysis, let us define diagonal matrices

$$D_{\mathbf{W}^{(0)}} = \mathbf{diag}\{\mathbb{1}_{\mathbf{W}^{(0)}(x,1)\geq 0}\} \qquad D_{\mathbf{V}^{(0)}, \mathbf{W}} = \mathbf{diag}\{\mathbb{1}_{\mathbf{V}^{(0)}(\mathsf{out}_1(x),1)\geq 0}\}$$

$$D_{\mathbf{W}} = \mathbf{diag}\{\mathbb{1}_{(\mathbf{W}^{(0)}+\mathbf{W})(x,1)\geq 0}\} \qquad D_{\mathbf{V}, \mathbf{W}} = \mathbf{diag}\{\mathbb{1}_{(\mathbf{V}^{(0)}+\mathbf{V})(\mathsf{out}_1(x),1)\geq 0}\}$$

which satisfy $\mathsf{out}_1(x) = \mathbf{A}D_{\mathbf{W}}(\mathbf{W}^{(0)} + \mathbf{W})(x, 1)$ and $\mathsf{out}(x) = \mathbf{A}D_{\mathbf{V}, \mathbf{W}}(\mathbf{V}^{(0)} + \mathbf{V})(\mathsf{out}_1(x), 1)$.

The proof of Theorem 1 can be divided into three simple steps with parameter choices in Table 1.

---

In this paper, we assume $0 < \alpha \leq \widetilde{O}(\frac{1}{kp_{\mathcal{G}}\mathfrak{C}_{\mathfrak{s}}(\mathcal{G})})$ and choose parameters

$$\sigma_w \in [m^{-1/2+0.01}, m^{-0.01}] \qquad\qquad \sigma_v \in [\mathsf{polylog}(m), m^{3/8-0.01}]$$

$$\tau_w := \widetilde{\Theta}(kp_{\mathcal{F}}\mathfrak{C}_{\mathfrak{s}}(\mathcal{F})) \geq 1 \qquad\qquad \tau_v := \widetilde{\Theta}(\alpha kp_{\mathcal{G}}\mathfrak{C}_{\mathfrak{s}}(\mathcal{G})) \leq \frac{1}{\mathsf{polylog}(m)}$$

and they satisfy $\quad \tau_w \in \left[m^{1/8+0.001}\sigma_w, m^{1/8-0.001}\sigma_w^{1/4}\right] \quad \tau_v \in \left[\sigma_v \cdot (k/m)^{3/8}, \frac{\sigma_v}{\mathsf{polylog}(m)}\right]$

---

Table 1: Three-layer ResNet parameter choices.
$\quad \sigma_w, \sigma_v$: recall entries of $\mathbf{W}^{(0)}$ and $\mathbf{V}^{(0)}$ are from $\mathcal{N}\left(0, \sigma_w^2\right)$ and $\mathcal{N}\left(0, \sigma_v^2/m\right)$.
$\quad \tau_w, \tau_v$: the proofs work with respect to $\|\mathbf{W}\|_2 \leq \tau_w$ and $\|\mathbf{V}\|_2 \leq \tau_v$.

In the first step, we prove that for all weight matrices not very far from random initialization (namely, all $\|\mathbf{W}\|_2 \leq \tau_w$ and $\|\mathbf{V}\|_2 \leq \tau_v$), many good "coupling properties" occur. This includes upper

---

kernel method.) Of course, as a simple corollary, our result also applies to training all the layers together, with appropriately chosen random initialization and learning rate.

[13]Performing SGD with respect to $\mathbf{W}^{(0)} + \mathbf{W}$ and $\mathbf{V}^{(0)} + \mathbf{V}$ is the *same* as that with respect to $\mathbf{W}$ and $\mathbf{V}$; we introduce $\mathbf{W}^{(0)}, \mathbf{V}^{(0)}$ notation for analysis purpose. Note also, one can alternatively consider having a training set and then performing SGD on this training set with multiple passes; similar results can be obtained.

bounds on the number of sign changes (i.e., on $\left\|D_{\mathbf{W}^{(0)}} - D_{\mathbf{W}}\right\|_0$ and $\left\|D_{\mathbf{V}^{(0)},\mathbf{W}} - D_{\mathbf{V},\mathbf{W}}\right\|_0$) as well as vanishing properties such as $\mathbf{A}D_{\mathbf{W}}\mathbf{W}^{(0)}, \mathbf{A}D_{\mathbf{V},\mathbf{W}}\mathbf{V}^{(0)}$ being negligible. We prove such properties using techniques from prior works [3, 5]. Details are in Section C.1.

In the second step, we prove the existence of $\mathbf{W}^*, \mathbf{V}^*$ with $\|\mathbf{W}^*\|_F \leq \frac{\tau_w}{10}$ and $\|\mathbf{V}^*\|_F \leq \frac{\tau_v}{10}$ satisfying $\mathbf{A}D_{\mathbf{W}^{(0)}}\mathbf{W}^*(x,1) \approx \mathcal{F}(x)$ and $\mathbf{A}D_{\mathbf{V}^{(0)},\mathbf{W}}\mathbf{V}^*(\mathsf{out}_1(x),1) \approx \alpha\mathcal{G}(\mathsf{out}_1(x))$. This existential proof relies on an "indicator to function" lemma from [3]; for the purpose of this paper we have to revise it to include a trainable bias term (or equivalently, to support vectors of the form $(x,1)$). Combining it with the aforementioned vanishing properties, we derive (details are in Section C.2):

$$\mathbf{A}D_{\mathbf{W}}\mathbf{W}^*(x,1) \approx \mathcal{F}(x) \quad \text{and} \quad \mathbf{A}D_{\mathbf{V},\mathbf{W}}\mathbf{V}^*(\mathsf{out}_1(x),1) \approx \alpha\mathcal{G}(\mathsf{out}_1(x)) \ . \quad (6.3)$$

In the third step, consider iteration $t$ of SGD with sample $(x_t, y_t) \sim \mathcal{D}$. For simplicity we assume $\mathsf{OPT} = 0$ so $y_t = \mathcal{H}(x_t)$. One can carefully write down gradient formula, and plug in (6.3) to derive

$$\Xi_t := \langle \nabla_{\mathbf{W},\mathbf{V}}\mathsf{Obj}(\mathbf{W}_t, \mathbf{V}_t; (x_t, y_t)), (\mathbf{W}_t - \mathbf{W}^*, \mathbf{V}_t - \mathbf{V}^*)) \rangle$$
$$\geq \tfrac{1}{2}\|\mathcal{H}(x_t) - \mathsf{out}(\mathbf{W}_t, \mathbf{V}_t; x_t)\|_2^2 - 2\|Err_t\|_2^2$$

with $\mathbb{E}\left[\|Err_t\|_2^2\right] \leq \widetilde{\Theta}\left(\alpha^4(kp_{\mathcal{G}}\mathfrak{C}_{\mathfrak{s}}(\mathcal{G}))^4\right)$. This quantity $\Xi_t$ is quite famous in classical mirror descent analysis: for appropriately chosen learning rates, $\Xi_t$ must converge to zero.[14] In other words, by concentration, SGD is capable of finding solutions $\mathbf{W}_t, \mathbf{V}_t$ so that the population risk $\|\mathcal{H}(x_t) - \mathsf{out}(\mathbf{W}_t, \mathbf{V}_t; x_t)\|_2^2$ is as small as $\mathbb{E}[\|Err_t\|_2^2]$. This is why we can obtain population risk $\widetilde{\Theta}\left(\alpha^4(kp_{\mathcal{G}}\mathfrak{C}_{\mathfrak{s}}(\mathcal{G}))^4\right)$ in (6.2). Details are in Section C.3 and C.4.

## 7 Overview of Theorem 3

Let us consider the following simple distribution of functions $\mathcal{H}(x) = \mathcal{F}(x) + \alpha\mathcal{G}(\mathcal{F}(x))$, with $\mathcal{F}(x) = \mathbf{W}^*x$ and $\mathcal{G}(y) = \left(\prod_{j \in [k]} y_j\right)_{i \in [k]}$. Here, $\mathbf{W}^* = \sqrt{d}(\mathbf{e}_{i_1}, \mathbf{e}_{i_2}, \cdots \mathbf{e}_{i_k})$ for $i_1 \neq i_2 \cdots \neq i_k$ being uniformly at random chosen from $[d]$.

**Theorem 2.** *For every constant $k \geq 4$, for sufficiently large $d$, for every $N \leq O(\frac{d^{k/2}}{\log^k d})$, for every $\alpha \in \left[\Omega(d^{-k/4}), 1\right)$, let $\mathcal{X} = \{x^{(1)}, \ldots, x^{(N)}\}$ be i.i.d. drawn from the uniform distribution over $\{\frac{\pm 1}{\sqrt{d}}\}^d$ and $y^{(i)} = \mathcal{H}(x^{(i)})$, and let $\mathfrak{K}(x)$ represent the optimal solution to (3.2) using any correlation kernel, then we have with probability at least 0.99 over the randomness of $\mathbf{W}^*$ and $\mathcal{X}$*

$$\mathbb{E}_{x \sim U(\{-1/\sqrt{d}, +1/\sqrt{d}\}^d)} \|\mathcal{H}(x) - \mathfrak{K}(x)\|_2^2 > \alpha^2/4$$

*Remark.* One can relax the assumption on $\mathfrak{K}(x)$ to being any *approximate* minimizer of (3.2).

In contrast, using Theorem 1, one can show $\mathfrak{C}_{\mathfrak{s}}(\mathcal{F}) = O(\sqrt{d})$ and $\mathfrak{C}_{\mathfrak{s}}(\mathcal{G}) = O(1)$ so SGD can find

$$\mathbb{E}_x \|\mathcal{H}(x) - \mathsf{out}(x)\|_2^2 \leq \widetilde{O}(\alpha^4) \quad \text{in} \quad N = \widetilde{\Theta}\left(\frac{d}{\alpha^8}\right) \text{ samples } .$$

As an example, when $k \geq 4$ and $\alpha = d^{-0.01}$, ResNet achieves regression error $\alpha^4$ in $N = \widetilde{O}(d^{1.08})$ samples, but kernel methods cannot achieve $\alpha^2$ error even with $N = \widetilde{\Theta}(d^{k/2}) = \widetilde{\Omega}(d^2)$ samples. We sketch the proof of Theorem 3 in half a page on Page 13.

**Conclusion.** We give the first provable separation between neural networks and kernel methods, in the efficient and distribution-free learning regime. We show that neural networks can implicitly hierarchical learn functions $\mathcal{G}(\mathcal{F}(x))$ with the help of $\mathcal{F}(x)$ using residual links, without paying sample complexity comparing to "one-shot" learning algorithms that directly learns $\mathcal{G}(\mathcal{F}(x))$. Finally, we would like to point out there are kernels not captured by our correlation kernels, such as convolutional networks with "global average pooling." To prove bounds for them is an interesting future direction.

## Footnotes

[2]Technically speaking, the three-layer learning theorem of [3] is beyond NTK, because the learned weights across different layers interact with each other, while in NTK the learned weights of each layer only interact with random weights of other layers. However, there exist other kernels — such as recursive kernels [39] — that can more or less efficiently learn the same concept class proposed in [3].

[3]Consider the class of degree-6 polynomials over 6 coordinates of the $d$-dimensional input. There exist two-layer networks with F-norm $O(\sqrt{d})$ implementing this function (thus have near-zero training and testing error). By Rademacher complexity, $O(d)$ samples suffice to learn if we are able to find a *minimal complexity* solution. Unfortunately, due to the non-convexity of the optimization landscape, two-layer networks can not be trained to match this F-norm even with $O(d^2)$ samples, see Figure 1.

[4] Of course, if one knew a priori the form $\mathcal{H}(x) = \langle w_1^*, x\rangle^{10} - \langle w_2^*, x\rangle^{10}$, one could also try to solve it directly by minimizing objective $(\langle w_1^*, x\rangle^{10} - \langle w_2^*, x\rangle^{10} + \langle w_2, x\rangle^{10} - \langle w_1, x\rangle^{10})^2$ over $w_1, w_2 \in \mathbb{R}^d$. Unfortunately, the underlying optimization process is highly non-convex and it remains unclear how to minimize it efficiently. Using matrix sensing [25], one can efficiently learn such $\mathcal{H}(x)$ in sample complexity $\widetilde{O}(d^5)$.

[5] However, this does not mean that the error of the first layer can be reduced by its own, since it is still possible for the first layer to learn $\mathcal{F} + \alpha \mathcal{R}(x) \pm \alpha^2$ and the second layer to learn $\mathcal{G}(\mathcal{F})(x) - \mathcal{R}(x) \pm \alpha$, for an arbitrary (bounded) function $\mathcal{R}$.

[6]In many cases, $R(w) = \lambda \cdot \sum_{j \in [k]} w_j^\top K_j w_j$ is the norm associated with the kernel, for matrix $K_j \in \mathbb{R}^{N \times N}$ defined as $[K_j]_{i,n} = K_j(x^{(i)}, x^{(n)})$.

[7]Recall a Mercer kernel $K \colon \mathbb{R}^{d \times d} \to \mathbb{R}$ can be written as $K(x,y) = \langle \phi(x), \phi(y) \rangle$ where $\phi \colon \mathbb{R}^d \to \mathcal{V}$ is a feature mapping to some inner product space $\mathcal{V}$.

[8]It is necessary the negative result of kernel methods is distribution dependent, since for trivial distributions where $x$ is non-zero only on the first constantly many coordinates, both neural networks and kernel methods can learn it with constantly many samples.

[9]If $R(w)$ is the $\ell_2$ regularizer, then this becomes a kernel method again since the minimizer can be written in the form (3.1). For other regularizers, this may not be the case.

[10]For general $\|w^*_{1,i}\|_2 \leq B, \|w^*_{2,i}\|_2 \leq B, |a^*_{r,i}| \leq B$, the scaling factor $B$ can be absorbed into the activation function $\phi'(x) = \phi(Bx)$. Our results then hold by replacing the complexity of $\phi$ with $\phi'$.

[11]Since we use ReLU networks as learners, they are positive homogeneous so to learn general functions $\mathcal{F}, \mathcal{G}$ which may not be positive homogenous, it is in some sense necessary that the inputs are scaled properly.

[12]This can be more meaningful than training all the layers together, in which if one is not careful with parameter choices, the training process can degenerate as if only the last layer is trained [11]. (That is a convex

[14]Indeed, one can show $\sum_{t=0}^{T-1} \Xi_t \leq O(\eta_w + \eta_v) \cdot T + \frac{\|\mathbf{W}^*\|_F^2}{\eta_w} + \frac{\|\mathbf{V}^*\|_F^2}{\eta_v}$, and thus the right hand side can be made $O(\sqrt{T})$ ignoring other factors.

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
