[Reviews · NeurIPS 2019]

Reviewer 1



Dear Authors: I read your rebuttal. I do indeed understand the point of your paper. I also agree that solving linear equations is not a good example of something kernel methods can't do. The 'isotonic regression' algorithm for learning a ReLU is a simple SGD algorithm that uses a straight-through estimator. The high-level message of your paper is that there are problems that gradient-based methods can solve, but kernel methods cannot. Learning a ReLU is a fine example. Your paper has many other interesting contributions such as working with vanilla SGD and also formally ruling out a variety of kernels. That's very nice, and I continue to recommend acceptance. The construction in this paper seems interesting and noteworthy. It is unfortunate that high level description of the lower bound is left until supplemental. Also, the paper leaves out many relevant works on this topic. I do not understand this submission's description of "Regularization Matters: Generalization and Optimization of Neural Nets vs their Induced Kernel." [37] does give a separation between neural nets and NTK (the submission here is better in the sense that they get a separation for all correlational kernels). Another recent paper by Yehudai and Shamir shows that a ReLU cannot be learned by random features. This implies that a ReLU cannot be learned by NTK, recursive kernel, etc. But a ReLU can be learned for all distributions by gradient descent on a surrogate (convex) loss see "Efficient Learning of Generalized Linear and Single Index Models with Isotonic Regression." So ReLU already seems to be a separation for many classes of kernels (though it is not proved for all correlational kernels). Building on this work "Learning Neural Networks with Two Nonlinear Layers" there is an algorithm that shows how to learn neural networks that cannot be embedded into known kernels (NTK, recursive) because they can learn ReLU, so again this goes beyond current kernel methods.

Reviewer 2



The results themselves form a nice study. The biggest problem I find is the impact. The studied network is neither convex nor concave but yet it is special (single-skip three-layer ResNet). It begs the question if the result can be extended to other networks, in particular without the skip connection, i.e., non ResNet. I do not see any practical relevance. The result says in layman words "use neural networks instead of kernel methods," which these days everybody takes as granted. Another big issues I have is with the exposition of the paper and the numerous grammar mistakes. The exposition of the paper is weird. It is impossible to read the paper without continuously going back and forth. For example, Theorem 1 is using complexity before it is actually formally defined. In the same theorem, probability is used but it is unclear what really is random (the samples are random but this is not the origin of this probability). I think a much better flow would be to define the quantities before being used. The two overview sections are kind of out-of-kilter. I'm not sure what a better flow would be but the current one definitely doesn't work well. My 'below accept' rate is due to unclear impact and relevance of the results, and very poor exposition of the work with numerous grammar mistakes. Minor comments (these are only a few of them; there are many additional grammar mistakes) 21: understood how 26: "from them" is incorrect 27: "to the optimization side" ??? 32: are kernel methods ... are defined 40: by random etc

Reviewer 3



(1) I have a serious concern about the parameter alpha: In Theorem 1, the prediction error delta needs to be larger than alpha^4. This means that if we require a high accuracy prediction, i.e., delta is small, alpha also needs to be small. Increasing the sample size/number of iterations T does not reduce th error. In another word, if we require the learnt network to be consistent, i.e., risk->0, then the composite signal in (2.2) needs to diminish. The scaling alpha->0 makes this regime not very reasonable. In contrast, if we consider alpha to be small fix constant, Theorem 1 will not give a meaningful bound. (2) The authors should highlight the difference between this paper and Allen-Zhu et al. 2018 for analyzing feedforward NN in terms of the proof techniques, as a large proportion of the proof technique in this paper are adapted from Allen-Zhu et al. 2018.

[Author Response · NeurIPS 2019]

We thank all the reviewers for the time reading our paper! We will fix all the minor issues, and below we only address
the main concerns. *We quickly point out in our revision we have already extended the lower bound to* all kernel methods,
*not just correlation kernels. We plan to include that if the paper gets in.*

- 4  **R2** thinks we might have left out some relevant works on this topic, by citing [37] as well as some learning theory
papers on ReLU: namely, "[L]earning Neural Networks with Two Nonlinear Layers" and "[E]fficient Learning of
Generalized Linear and Single Index Models with Isotonic Regression".

We're afraid R2 has misunderstood our work. Our main contribution is to show "can **Neural Networks** learn some
concept class in distribution-free, efficiently-learnable setting while kernel methods can not."

   - 9  ReLU functions, in particular, are **not** known to be distribution-freely (efficient) learnable by **standard ReLU**
**neural network**. In the cited prior works [L] and [E], the main learning algorithms are both isotonic regression,
which is **not training a neural network** via SGD. Generally, our work is not about "there exist learning methods
better than kernels", which is trivially true (in fact, linear regression using Gaussian elimination is not doable by
kernel methods). Our main contribution is to show that "Neural Networks can be better learners than kernels (esp.
NTK)". This was not known at all in the distribution-free setting, which we have emphasized in the introduction.
   - 15  For [37], its separation works *only when* the solution (i.e., the network weights) corresponds to a "max-margin
solution" (which means minimal-norm solution in our language). This is *not* necessarily efficiently learnable.
We have made this point clear on lines 50-60 and given a counter example in Fig 1.

- 18  **R2:** give more explanation for lower-bound (LB) construction, and can it be extended to output dimension 1?

Due to space limitation, we didn't explain much about LB. Sorry. Our proof overview is short and on page 12, line
371-379. Intuitively, we constructed a "high-complexity polynomial" $G(F(x))$ that is degree $k$ over $d$ Boolean
inputs. By Boolean analysis, we write the solution of kernel method in the Boolean Fourier basis, and then argue
about its certain coefficients. (If we can learn $G(F(x))$ then some coefficient must be large, but it is too large so
that the kernel method will memorize training data unless $d^{k/2}$ samples are given).

Our lower bound can indeed be extended to networks with one dimension output. We choose $k \geq 2$ to present the
simplest proof.

- 26  **R4:** can result be extended to networks without skip connection? Our experiment suggests that non-ResNet cannot
learn this concept class with comparable sample complexity, so having skip connection is helpful in our setting.

- 28  **R4:** What's the practical relevance? It says "neural networks are better than kernels", but that's known in practice.

While known in practice, we provide a *formal prove*. As we emphasized in the intro, perhaps to many practitioner's
surprise, there was no supporting argument (in theory) before this work for "neural networks are better learners than
kernels" at least in distribution-free (efficiently computable) setting. This is the setting most relevant to practice.
This is also why our theoretical contribution is significant.

- 33  **R4** has given suggestions regarding reformatting this paper. Thank you and we will keep this in mind.

- 34  **R5** has concern about $\alpha^4$ and worries that the risk (i.e. error) cannot go below $\alpha^4$

Our setting is different from traditional statistical learning, where the risk can go to 0 for a fixed concept class.
Instead, for every $risk > 0$, we construct a different concept class. The correct way to interpret our paper is that

"for every $\alpha > 0$ (which defines $H$ and a concept class), NN can learn up to accuracy $\alpha^4$, but NTK cannot learn up
to accuracy $\alpha^2$, if the same sample size (around $poly(1/\alpha)$) is given to both methods."

Furthermore, Theorem 1 does give a meaningful bound when $\alpha$ is very small, since the constants in $O(\cdot)$ in Theorem
1 is *universal* and does not depend on $\alpha$. Hence, ResNet can learn the target class with non-trivial accuracy with
relatively few samples (think of a function with output around 1, then learning the function up to error 0.1 is very
meaningful, although not asymptotic). More importantly, with these many samples, the error is *significantly smaller*
than the best solution obtained by kernel methods. Our setting is not asymptotic, but still commonly considered in
machine learning.

Finally, we admit that supporting risk$\rightarrow 0$ (for a fixed concept class) would be an important future direction.

- 46  **R5:** what's the overlap with Allen-Zhu et al. 2018?

*Short answer:* lower bound has no overlap; for upper bound, Allen-Zhu et al [2][3] are essentially kernel methods;
since we have proven lower bound for kernels, [2][3] must fail to produce $\alpha^4$ error as we do in Theorem 1.
*Long answer:* [2][3] focus on the setting where (more or less) the sign patterns of the ReLU's do not change during
training. Hence, the network training process is similar to the NTK kernel regime (see line 35-42, as well as criticism
in arXiv 1904.00687). In this paper, we show that ResNet can learn functions that are not learnable by NTK. This is
due to the fact that the sign patterns change a lot, which requires new techniques. (Of course, to make this paper
short, in proving upper bound we have adopted some known lemmas from [2].)

[Meta-Review · NeurIPS 2019]

There has been many theoretical papers studying neural nets in the setting that they behave like kernels. This work shows a clear example of functions that 1) cannot be learned in kernel setting 2) a neural net can learn it efficiently. Even though limitations of kernel methods are known among practitioners, this result is significant as it characterizes these limitations in a provable way. Therefore, I recommend acceptance.